# Comparison of Machine Learning Techniques for Prediction of Hospitalization in Heart Failure Patients

**DOI:** 10.3390/jcm8091298

**Published:** 2019-08-24

**Authors:** Giulia Lorenzoni, Stefano Santo Sabato, Corrado Lanera, Daniele Bottigliengo, Clara Minto, Honoria Ocagli, Paola De Paolis, Dario Gregori, Sabino Iliceto, Franco Pisanò

**Affiliations:** 1Unit of Biostatistics, Epidemiology and Public Health, Department of Cardiac, Thoracic, Vascular Sciences and Public Health, University of Padova, Via Loredan, 18, 35131 Padova, Italy; 2MediaSoft, via Sonzini, 25, 73013 Galatina (Le), Italy; 3AUSL/Lecce, Zona Draghi, 73039 Tricase (Le), Italy; 4Cardiology Unit, Department of Cardiac, Thoracic, Vascular Sciences and Public Health, University of Padova, Via Giustiniani, 2, 35128 Padova, Italy

**Keywords:** heart failure, machine learning techniques, hospitalization

## Abstract

The present study aims to compare the performance of eight Machine Learning Techniques (MLTs) in the prediction of hospitalization among patients with heart failure, using data from the Gestione Integrata dello Scompenso Cardiaco (GISC) study. The GISC project is an ongoing study that takes place in the region of Puglia, Southern Italy. Patients with a diagnosis of heart failure are enrolled in a long-term assistance program that includes the adoption of an online platform for data sharing between general practitioners and cardiologists working in hospitals and community health districts. Logistic regression, generalized linear model net (GLMN), classification and regression tree, random forest, adaboost, logitboost, support vector machine, and neural networks were applied to evaluate the feasibility of such techniques in predicting hospitalization of 380 patients enrolled in the GISC study, using data about demographic characteristics, medical history, and clinical characteristics of each patient. The MLTs were compared both without and with missing data imputation. Overall, models trained without missing data imputation showed higher predictive performances. The GLMN showed better performance in predicting hospitalization than the other MLTs, with an average accuracy, positive predictive value and negative predictive value of 81.2%, 87.5%, and 75%, respectively. Present findings suggest that MLTs may represent a promising opportunity to predict hospital admission of heart failure patients by exploiting health care information generated by the contact of such patients with the health care system.

## 1. Introduction

The sudden development of health technologies fostered the opportunity of measuring a large amount of clinical data with the final aim to improve patients’ management [1]. Nowadays, physicians and medical researchers can constantly monitor clinical data of each patient, allowing for accurate tracking of the disease’s evolution. Such data are generally collected and stored in electronic health records (EHR), which holds promise to improve efficiency and quality of healthcare, making data more accessible, facilitating health information exchange and interoperability between healthcare providers [2]. The benefits of EHR technology are even more relevant in chronic diseases, such as cardiovascular disorders, where a lifelong disease management is crucial to avoid disease relapse (and its consequences, including, but not limited to, hospital readmissions, high cost of care, and premature mortality), which represents a severe public health burden [3].

Heart failure (HF) represents a clear example of chronic cardiovascular disease requiring lifelong management [4,5]. It is strongly related to the aging process [6], and it is associated with high healthcare resource utilization [7] so that the improvement of HF management should be one of the primary goals of current health organizations [8].

Stratifying HF patients according to their risk of disease relapse and consequent hospital admission would be useful from both the clinical and economic standpoints. Identifying those HF patients at high risk of hospital admission would be useful for the clinicians since they could focus on the management of such patients to prevent potential disease relapse. From the point of view of health care planning, this information would be useful in the allocation of economic resources. However, even though the availability of a large amount of data would be a great opportunity to characterize better patients suffering from chronic disease, it has been shown that it is extremely difficult to transform such complex information in useful knowledge.

Machine learning techniques (MLTs) offer a new possibility in terms of the management of this information. A growing body of literature shows MLT applications in cardiology, especially for developing prediction models using both supervised and unsupervised methods [9]. In recent years, MLTs have been increasingly used also in the field of HF research [10,11]. Those fields most frequently investigated using MLTs are the identification and classification of HF cases, prediction of HF treatment adherence, prediction of HF-related adverse events, and prediction of hospital admission/readmissions of HF patients [10,11]. Prediction of hospital admission/readmission of HF patients and, in general, heart disease patients, is of great interest given the healthcare resource burden related to hospital admission/readmission. A recent study [12] showed an outperformance of random forest (RF) compared to traditional logistic and Poisson regression in predicting HF readmissions. Conversely, another study [13] showed no improvements using MLTs (RF, tree-augmented naive Bayesian network, and a gradient-boosted model) compared to traditional techniques in predicting hospital readmissions of HF patients. For what concerns specifically hospital admissions, a study compared five different techniques (support vector machine (SVM), adaboost (AB), naive Bayes, K-likelihood ratio test, logistic regression (LR)) and showed similar predictive performances [14]. Even though MLTs seem to be a promising opportunity to predict hospital admission/readmission in HF patients, literature results are still inconsistent. 

The present study aims to compare the performance of several MLTs in the prediction of hospitalization among patients with HF, using data from the Gestione Integrata dello Scompenso Cardiaco (GISC) study [15]. We compared the following algorithms: LR, generalized linear model net (GLMN), classification and regression tree (CART), logitboost (LB), AB, RF, SVM, and neural network (NN).

## 2. Materials and methods

### 2.1. Gestione Integrata dello Scompenso Cardiaco (GISC) Study

Data analyzed in the present study have been derived from the GISC study. It is an ongoing project, and it takes place in the region of Puglia, Southern Italy [15]. Patients with a diagnosis of HF are enrolled in a long-term assistance program that includes the adoption of an online platform for data-sharing between general practitioners and cardiologists working in hospitals and community health districts. The diagnosis of HF is made according to the criteria of the European Society of Cardiology (ESC) [5] using the patient’s clinical history, physical examination, and data obtained from clinical tests (electrocardiogram, echocardiography, N-terminal pro-brain natriuretic peptide (NT-proBNP)). 

This informative database includes patients’ demographic and clinical information (anthropometric characteristics, etiology of HF, presence of comorbidities, results of blood examination, and number of hospitalizations). Data are collected by cardiologists and family doctors involved in the primary care of patients with HF. 

The primary outcome of the present study is to compare the performance of eight machine learning techniques in the prediction of hospitalization of patients with HF enrolled in the GISC project. The secondary outcome is to identify predictors of hospitalization in such patients.

For the study, we analyzed data of 380 HF patients (with both preserved and reduced ejection fraction (EF)) enrolled between 2011 and 2015. Patients were distributed as follows: 21% (reduced EF), 55% (mid-range EF), 24% (preserved EF). No standard follow-up schedule has been foreseen in this study. In general, the follow-up was at least once a year, but, depending on clinical conditions and medical judgement, it could be more frequent.

Out of 380 records, 110 had no missing data (complete cases). Among the 380 patients, 170 were not hospitalized, and 210 had at least one hospital admission. The following patient’s characteristics were considered in the analysis as potential predictors of hospitalization:Numerical variables: body mass index (BMI), age, heart rate, BNP, pulmonary pressure, serum creatinine, mean years between clinical examinations at follow-up;Categorical variables: gender, the occurrence of myocardial infarction, etiology related to ischemic cardiomyopathy, dilated cardiomyopathy or valvulopathy, presence of comorbidities, chronic obstructive pulmonary disease (COPD) or anemia (dichotomous data), and New York Heart Association (NYHA) class (ordinal data).

Descriptive statistics were reported as I quartile/median/III quartile for continuous variables and percentages (absolute numbers) for categorical variables. A Wilcoxon–Kruskal–Wallis test was performed for continuous variables and a Pearson chi-square test for categorical ones.

We transformed all the predictors into numerical variables, and we created dummy variables for categorical predictors. All the continuous variables were rescaled into the range −1 and 1 and centered on the mean [16]. 

### 2.2. Machine Learning Techniques

LR, GLMN, CART, RF, LB, AB, SVM and NN were applied to evaluate the feasibility of such techniques in predicting hospitalization of patients with HF. We decided to compare these algorithms, given their increasing popularity in clinical settings for prediction of binary outcomes and their ability to detect complex relationships between the outcome and predictors and interactions between covariates [17,18].

LR is perhaps the method most frequently used to predict the occurrence of an event in clinical research [19]. The popularity of LR is mainly related to its ability to provide meaningful and easy-to-interpret quantities such as odds ratios (ORs), which can provide clinical information on the impact of predictors on the occurrence of the event of interest. However, LR is known to have some limitations given its parametric assumptions and the difficulty to detect non-linearities and interactions between covariates. LR was often used as a benchmark in studies aimed to compare different MLTs for the prediction of the occurrence of a binary outcome [20,21].

GLMN is a regularized regression model computed to linearly combined lasso and ridge penalties (L1 and L2) with a link function and a variance function to reduce linear model limitations [22]. GLMN is a technique that is often used in prediction settings where the researcher is interested in the identification of a subset of covariates that are strong predictors of the outcome of interest. This model works very well with data characterized by high collinearity among covariates [23].

CART and RF are tree-based techniques. CART is a technique that builds a simple decision tree on the analyzed data [24]. It uses a recursive binary splitting algorithm to divide the space of the predictors. After that, predictions are carried out in each region formed by the binary splitting. CARTs are becoming very popular in clinical settings because they are simple to implement and easy to interpret [18,25,26]. Despite their simplicity, CARTs often suffer from overfitting problems, which can often undermine their predictive reliability [23]. RF is an extension of CART. It works by constructing one CART on several bootstrap replicates of the original data [27]. In addition to that, it allows us to build each tree using only a subset of the available potential predictors. The final predictions are then obtained by averaging the predictions of each tree. RF has been shown to perform very well in several medical settings [28,29].

AB and LB are boosting algorithms that aim to combine several weak classifiers with improving classification performance [30]. Each weak classifier is implemented using decision trees with one single split. Each learned classifier is then combined in a weighted sum that returns the final boosted algorithm. The LB algorithm is a generalization of the AB, i.e., the AB algorithm is considered as a generalized additive model with binomial family and the logit link function [31]. These techniques have been shown to have good predictive performances in many clinical applications [28,32,33].

SVM is an algorithm that was developed for binary classification settings with two classes [34]. SVM works by constructing hyperplanes of the covariates’ space that separates the observations according to the class they belong to. The separation is carried out by augmenting the features’ space using kernel functions to allow for non-linear relationships between the outcome and the covariates. The use of such kernel functions allows the analysts to detect and model complex relationships, which can be very common in clinical research. SVM showed good classification ability in several settings, and it has been proven to be a good competitor of other MLTs [35,36,37].

NNs are a generalization of linear regression functions [38]. NNs are characterized by units, called neurons, which are connected. In its simplest form, the neurons take the information from the input units, i.e., the value of the predictors in the dataset, computed a weighted sum of the received inputs and provide an output, which, in classification tasks, is the class predicted by the NN for each observation. NNs are implemented using many parameters such that they can flexibly approximate any smooth functions. NNs have been widely used in pattern recognition field [39,40,41] and they have recently become very popular in medical research, being shown to outperform many other MTLs [42,43,44].

### 2.3. Model Training and Testing

The goal of the analysis was to compare MLTs in terms of ability to correctly classify patients that had at least one hospital admission and not to model time to the hospitalization. The study aims to understand how MLTs can enhance the classification of hospitalizations in the defined period, i.e., five years, and this has been shown to be a more sensitive question, noticeably in an MLT context than other traditional approaches in modeling long-term events and mortality [45]. 

Model tuning and validation were carried out using a 5-fold cross-validation approach [23] on all the patients available in the dataset. For each method, the optimal parameters values were chosen with a grid search approach such that the cross-validated accuracy (the average proportion of correctly classified observations across the 5 folds) was maximized. The model with the optimal parameters was chosen as the final model to be compared with the others. The predictive abilities of MLTs were assessed using the following measures: positive predictive value (PPV), negative predictive value (NPV), sensitivity, specificity, accuracy, and area under the ROC curve (AUC). Each measure was computed, averaging the value obtained on each resampling fold. We computed the Cohen’s Kappa statistics of agreement [46], with their corresponding 95% confidence intervals (CIs), to measure the degree of concordance of each pair of techniques in the predicted class.

Three different approaches were explored to handle missing data: Complete Case (CC) analysis, i.e., only the patients with complete information for all the variables were included in the analysis, imputation of missing values with median values for numerical variables and the most frequent class for categorical variables, and imputation of missing data with K-nearest neighbors (KNN) algorithm [47]. We will refer to these three approaches as CC, Median Imputation (M-I), and KNN imputation (KNN-I). The imputation of missing data was implemented during the validation process, as it was shown to provide more reliable insights on the predictive ability of the models [48]. We compared the performances of MLTs under all the approaches to test the sensitivity of each method to different strategies for handling missing data.

A power analysis with a simulation-based approach was run to evaluate the minimum sample size needed to properly train the MLTs [49]. A logistic regression was assumed to describe the association between predictors and the presence of at least one hospital admission, assuming to estimate an AUC of 70%, in line with previous findings [12,13], with a 10% margin of error and a percentage of patients with at least one hospital admission of nearly 50% [50]. A minimum number of 340 records was identified.

All the analyses were implemented using the R Statistical Software [51] (version 3.6.0) with the following packages: *glmnet* [52] for the GLMN algorithm, *rpart* [53] for the CART algorithm, *ranger* [54] for RF, *caTools* [55] for LB, *adabag* [56] for the AB algorithm, *e1071* [57] for the SVM algorithm, and *nnet* [58] for NNs. Model tuning and validation was performed using *caret* [59] package, the *tidyverse* bundle of packages [60] was used for data management, functional programming and plots.

## 3. Results

The analyses considered 380 cases. The median duration of the follow-up was 1184 days (I quartile: 821 and III quartile: 1682). The distribution of the sample characteristics is reported in Table 1. Two hundred and ten patients were hospitalized, the distribution of the hospitalizations according to the cause of hospital admission was as follows: 84.76% HF, 15.24% other causes. In the sample there was a high proportion of COPD patients, especially among those hospitalized Significant differences were observed also for anemia prevalence (higher in the patients hospitalized, *p*-value 0.045). In addition to that, also, creatinine levels and BNP were higher in patients hospitalized (*p*-value 0.021 and < 0.001, respectively). 

Overall, 270 records (71% of the subjects) showed missing information in at least one of the variables. BMI and pulmonary pressure showed the highest percentages of missing values, i.e., 49% and 42% respectively. Age, NYHA class, creatinine level, heart rate and BNP had percentages of missing values between 3% and 5%. All the other variables had no missing information.

Table 2, Table 3 and Table 4 show the predictive performances of MLTs with the CC, M-I, and KNN-I approaches, respectively. Predictive performances were higher when all the patients with missing information for at least one variable were removed from the analysis. GLMN outperforms the other MLTs among those implemented with CC analysis, with higher values of all the measures used to compare the algorithms.

Agreement between MLTs’ predictions was assessed on MLTs obtained with CC analysis since it was the approach that returned the highest predictive performances. Cohen’s Kappa estimates, along with their 95% CIs, between each pair of MLTs are shown in Table 5. Overall, all the techniques showed moderate agreement. The highest indexes values were observed for the pair GLMN–LB, SVM–LR, NN–RF, AB–SVM, and AB–LR, which show an almost perfect agreement between predicted classes.

We evaluated the impact of predictors in identifying patients that had at least one hospitalization using GLM, LR, CART and RF trained with CC, i.e., the approach that showed the best performance. Regarding GLMN, predictors that had a coefficient different from zero were identified as having a predictive value. Among them, predictors were considered as “important” if the likelihood ratio test showed a *p*-value less than 0.05 for LR, whereas covariates that reduced the predictive error of the models with permutation methods were labelled as important for CART and RF [61]. Table 6 shows which covariates were identified to have an impact on identifying patients with at least a hospital admission. Had suffered from acute myocardial infarction (AMI), ischemic cardiomyopathy and suffering from comorbidities were identified as important predictors by all the four MLTs.

GLMN with CC analysis, i.e., the model found to have the best performance, was re-fitted on 10,000 bootstrap resampling to estimate coefficients’ distributions (median and 95% CI) to understand the impact that each predictor has in the model performance. ORs’ distributions (median and 95% CI) for 10,000 bootstrap repetitions for each one of the variables included in the model are shown in Table 7. Suffering from comorbidities, higher levels of creatinine and pulmonary pressure, and had suffered from AMI and ischemic cardiomyopathy were found to be significantly associated with a higher risk of hospitalization.

## 4. Discussion

The present study aimed to compare the performance, in terms of accuracy level, of different MLTs in predicting hospital admission of patients with HF enrolled in the GISC study. The GLMN was found to have the best performance in predicting the hospitalization, with an average accuracy, positive predictive value and negative predictive value of 81.2%, 87.5%, and 75%, respectively, even though the performance of the other MLTs was quite poor. From the clinical point of view, MLTs represent a promising opportunity to develop models able to predict hospital admission/readmission of HF patients based on data characterized by complex relationships and non-linear interactions. Not least, in the long run, we can expect that the predictive models will help the clinicians by identifying specific profile of patients (in terms of clinical characteristics) at risk of hospital admission. 

The present work showed that, with the exception of GLMN, the predictive performance of the MLTs was quite poor. However, we cannot make conclusions about the usefulness of such methods in developing predictive models using clinical data. We cannot rule out that using a larger database and/or more detailed clinical information would improve the predictive performance, especially if the clinical data employed to predict the hospitalizations are characterized by complex relationships and non-linear interactions. Nevertheless, in such settings and the limited information available, it is not uncommon to don’t observe an overperformance of logistic regression as compared to other MLTs [62]. The work of Frizzel JD. et al. [13] did not find out an outperformance of MLTs compared to more traditional techniques in predicting hospital readmission of HF patients, and that of Dai W. et al. [14] showed a similar performance of MLTs, including logistic regression, in predicting hospitalizations of heart disease patients. Choosing on a priori basis the model which is most likely to be more appropriate is not an easy task. Some guidance has been reported in literature [63], and we largely followed this in approaching this work. A re-arranged synthesis of the main characteristics of the algorithm has been reported as Appendix A.

As concerns hospitalization predictors, surprisingly, the BNP and the NYHA class were found to be predict hospital admission only by the RF and the CART approaches. Such results could be related to the sample characteristics—which were homogeneous in terms of NYHA class since about two-thirds of the patients had an NYHA class of 3—or to the sample size which could be not enough to identify such characteristics as significant predictors of hospitalization.

It is worth pointing out that it is difficult to compare results from different studies that employ MLTs to predict hospital admission/readmission in HF patients. Each study employed a different type of information, including only clinical data collected in the context of previous hospital admissions (when hospital readmission is predicted), health claims data, and both clinical and administrative data. Undoubtedly, the added value of the present study is that the data have been derived from a study in which clinical information is collected beyond the hospital setting (since not only hospital cardiologists, but also community health district cardiologists and general practitioners are involved in the data collection). This is crucial to better characterize individual health status. We cannot restrict our analysis to information collected during a single event of interest (e.g., the hospitalization). The individual medical history is a constant flow of information related to every single aspect of a patient’s life. Considering such a framework, it becomes even clearer the relevance of adopting appropriate data analysis techniques to exploit such complex information. The exploitation of such complex information is crucial to improve patients’ clinical management but also related costs. In the case of HF, an overall cost of more than $100 billions per year (including both direct and indirect costs) has been estimated [64]. Application of MLTs can substantially contribute to the creation and dissemination of new knowledge, and thus improving the planning of health care services cost-effectively [65] (e.g., by concentrating the resources on HF patients at high risk of hospitalization to avoid disease relapse). 

The main limitations of the present study are represented by the low number of records used to train the models and the lack of an external validation dataset. Such aspects could lead to the risk of overfitting and, consequently, undermine the reliability of the algorithms; further data should be collected to improve models’ implementation. Since the data collection of the GISC study is ongoing, we expect to improve further the implementation of MLTs to predict hospital admission of such patients. 

## 5. Conclusions

Present findings suggest that MLTs may be a promising opportunity to predict hospital admission of HF patients by exploiting health care information generated by the contact of such patients with the health care system, in the context of the GISC study. However, further research is needed to improve their accuracy level and to better evaluate their usefulness in clinical practice.

## Figures and Tables

**Table 1 jcm-08-01298-t001:** Sample characteristics. Continuous data are reported as I quartile/Median/III quartile, categorical data are reported as percentage (absolute number).

	Not Hospitalized (*N* = 170)	Hospitalized (*N* = 210)	*p*-Value
Gender: Female	54% (92)	60% (125)	0.29
Age	72.0/78.0/83.0	73.0/79.0/83.0	0.357
BMI	25.78/29.33/33.21	25.49/29.37/34.75	0.99
**Medical history**			
AMI	12% (21)	12% (26)	0.993
HF etiology—ischemic cardiomyopathy	15% (25)	22% (47)	0.058
HF etiology—dilated cardiomyopathy	9% (16)	10% (21)	0.847
HF etiology—valvulopathy	18% (30)	21% (45)	0.357
COPD	26% (45)	45% (94)	<0.001
Anemia	15% (25)	23% (48)	0.045
Comorbidities	39% (67)	48% (101)	0.09
**Clinical examination**			
Heart rate	75.0/90.0/100.0	80.0/90.0/94.25	0.098
BNP	850/1335/3000	1178/2228/3680	<0.001
Pulmonary pressure	35/40/47	35/41.5/52	0.051
NYHA class			0.914
2	24% (39)	26% (53)	
3	67% (107)	66% (136)	
4	9% (14)	8% (16)	
Creatinine	0.800/1.000/1.208	0.810/1.070/1.450	0.021
Mean years between clinical examinations	0.625/1.600/2.900	0.900/1.800/2.900	0.281

BMI: body mass index; AMI: acute myocardial infarction; HF: heart failure; COPD: chronic obstructive pulmonary disease; BNP: beta-type natriuretic peptide; NYHA: New York Heart Association.

**Table 2 jcm-08-01298-t002:** Performance of generalized linear model net (GLMN), logistic regression (LR), classification and regression tree (CART), random forest (RF), adaboost (AB), logitboost (LB), support vector machine (SVM), and neural network (NN) obtained with complete case (CC) analysis. The values represent sensitivity, positive predictive value (PPV), negative predictive value (NPV), specificity and accuracy averaged over the values obtained on each resample.

Technique	Sensitivity	PPV	NPV	Specificity	Accuracy	AUC
**GLMN**	77.8	87.5	75	85.7	81.2	80.6
**LR**	54.7	51.6	64.9	61.9	58.9	64.6
**CART**	44.3	61.6	65.4	78.1	63.5	58.6
**RF**	54.9	73.0	72.7	85.6	72.6	69.1
**AB**	57.3	63.8	70.8	74.4	67.1	64.4
**LB**	66.7	66.7	57.1	51.1	62.5	65.4
**SVM**	57.3	69.0	72.2	79.4	69.9	69.5
**NN**	61.6	62.8	72.4	73.1	68.2	67.7

**Table 3 jcm-08-01298-t003:** Performance of GLMN, LR, CART, RF, AB, LB, SVM, and NN obtained with M-I analysis. The values represent sensitivity, positive predictive value (PPV), negative predictive value (NPV), specificity and accuracy averaged over the values obtained on each resample.

Technique	Sensitivity	PPV	NPV	Specificity	Accuracy	AUC
**GLMN**	26.5	66.0	59.5	68.1	60.3	62.8
**LR**	54.7	57.9	65.2	68.1	62.1	64.1
**CART**	40.0	56.6	60.9	74.3	58.9	57.2
**RF**	50.6	64.2	65.7	76.7	65.0	66.7
**AB**	56.5	62.1	67.5	72.4	65.3	68.0
**LB**	50.0	61.2	64.8	72.5	62.5	58.9
**SVM**	66.5	57.7	69.2	60.5	63.2	63.6
**NN**	28.8	58.2	59.1	83.3	58.9	61.9

**Table 4 jcm-08-01298-t004:** Performance of GLMN, LR, CART, RF, AB, LB, SVM, and NN obtained with KNN-I analysis. The values represent sensitivity, positive predictive value (PPV), negative predictive value (NPV), specificity and accuracy averaged over the values obtained on each resample.

Technique	Sensitivity	PPV	NPV	Specificity	Accuracy	AUC
**GLMN**	24.1	64.8	59.4	89.5	60.3	62.4
**LR**	54.1	57.6	64.9	68.1	61.8	63.2
**CART**	45.3	54.4	61.2	69.5	58.7	57.8
**RF**	50.6	64.2	65.7	76.7	65.0	66.7
**AB**	53.5	60.2	65.3	71.0	63.2	65.4
**LB**	60.7	60.3	68.8	67.9	65.0	64.2
**SVM**	53.5	57.2	64.3	67.6	61.3	62.2
**NN**	55.9	58.5	65.9	68.1	62.6	64.1

**Table 5 jcm-08-01298-t005:** Agreement between the class predicted by pair of machine learning techniques (MLTs) with complete case (CC) analysis. The values represent the point estimates of Cohen’s Kappa index along with their 95% CIs.

	NN	LB	SVM	LR	AB	CART	RF
**GLMN**	0.8 (0.64–0.95)	1 (1–1)	0.75 (0.59–0.91)	0.75 (0.59–0.91)	0.75 (0.59–0.91)	0.8 (0.65–0.95)	0.77 (0.61–0.93)
**NN**	_	0.77 (0.61–0.93)	0.51 (0.35–0.68)	0.51 (0.35–0.68)	0.51 (0.35–0.68)	0.92 (0.85–1)	1 (1–1)
**LB**	_	_	0.54 (0.38–0.7)	0.54 (0.38–0.7)	0.54 (0.38–0.7)	0.73 (0.6–0.86)	0.69 (0.55–0.83)
**SVM**	_	_	_	1 (1–1)	1 (1–1)	0.55 (0.39–0.71)	0.51 (0.35–0.68)
**LR**	_	_	_	_	1 (1–1)	0.55 (0.39–0.71)	0.51 (0.35–0.68)
**AB**	_	_	_	_	_	0.55 (0.39–0.71)	0.51 (0.35–0.68)
**CART**	_	_	_	_	_	_	0.92 (0.85–1)

**Table 6 jcm-08-01298-t006:** Covariates identified by the MLTs trained with CC to have predictive value in identifying patients that had a hospitalization. The symbol “X” denotes that the covariate had predictive value, whereas an empty cell denotes that the covariate had no predictive value. The symbol “_” was used for the MLTs for which it was not possible to identify covariates that had a predictive impact.

	GLMN	LR	CART	RF	AB	LB	SVM	NN
Gender (female vs. male)					_	_	_	_
Age					_	_	_	_
BMI					_	_	_	_
**Medical history**					_	_	_	_
AMI (yes vs. no)	X	X	X	X	_	_	_	_
HF etiology–ischemic cardiomyopathy (yes vs. no)	X	X	X	X	_	_	_	_
HF etiology–dilated cardiomyopathy (yes vs. no)					_	_	_	_
HF etiology–valvulopathy (yes vs. no)					_	_	_	_
COPD (yes vs. no)	X			X	_	_	_	_
Anemia (yes vs. no)				X	_	_	_	_
Comorbidities (yes vs. no)	X	X	X	X	_	_	_	_
**Clinical examination**					_	_	_	_
Heart rate				X	_	_	_	_
BNP			X	X	_	_	_	_
Pulmonary pressure	X		X	X	_	_	_	_
NYHA class				X	_	_	_	_
Creatinine	X		X	X	_	_	_	_
Mean years between clinical examinations		X	X	X	_	_	_	_

BMI: body mass index; AMI: acute myocardial infarction; HF: heart failure; COPD: chronic obstructive pulmonary disease; BNP: beta-type natriuretic peptide; NYHA: New York Heart Association.

**Table 7 jcm-08-01298-t007:** Coefficients’ distributions (median and 95% CI) for 10,000 bootstrap repetitions of the model found to have the best performance, i.e., GLMN (alpha = 0.005, lambda = 1/6).

	95% CI lower limit	Median	95% CI upper limit
Gender (female vs. male)	0.80	0.98	1.19
Age	0.99	1	1.02
BMI	0.98	1	1.01
**Medical history**			
AMI (yes vs. no)	1.08	1.41	1.74
HF etiology—ischemic cardiomyopathy (yes vs. no)	1.05	1.31	1.57
HF etiology—dilated cardiomyopathy (yes vs. no)	0.73	1	1.36
HF etiology—valvulopathy (yes vs. no)	0.71	0.90	1.15
COPD (yes vs. no)	1	1.22	1.49
Anemia (yes vs. no)	0.96	1.19	1.40
Comorbidities (yes vs. no)	1.12	1.34	1.44
**Clinical examination**			
Heart rate	0.99	1	1
BNP	1	1	1
Pulmonary pressure	1	1.01	1.02
NYHA class	0.72	0.91	1.14
Creatinine	1.01	1.21	1.40
Mean years between clinical examinations	0.99	1.08	1.17

BMI: body mass index; AMI: acute myocardial infarction; HF: heart failure; COPD: chronic obstructive pulmonary disease; BNP: beta-type natriuretic peptide; NYHA: New York Heart Association.

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
