# Peer review of "Comparison of Machine Learning Techniques for Prediction of Hospitalization in Heart Failure Patients"

_jcm, 2019, doi:10.3390/jcm8091298_

Round 1
Reviewer 1 Report
Please add a table comparing your methods with those reported in the literature.
Author Response
Choosing on a-priori basis the model which most likely can be more appropriate, it is not an easy task. Some guidance have been reported in literature, and we largely followed them in approaching this work. A re-arranged synthesis of the main characteristics of the algorithm has been reported as Supplementary Material.
Reviewer 2 Report
The investigators compared different methods of machine learning to analyse data collected routinely from hospital and community-care settings, concerning patients diagnosed to have heart failure, for their predictive power to identify patients who will be readmitted.
The number of subjects is relatively small for this type of study, at 388. Some data were missing in 71% of patients, so for some analyses only those patients with full data (N=110) were used. Perhaps to compensate for the limited number, analyses using the best performing method (GLMNet) were repeated 10,000 times, with a bootstrapping method of resampling, and then mean values for all the iterations were reported. Obviously, that risks overfitting, as the authors identify, and they do not report the performance of their method in a retest population that is independent.
The main concerns are that the paper misses important details. There is no information about how heart failure was diagnosed, and whether the population includes both heart failure with reduced and with preserved ejection fraction. It is implied that the aim of the study is to follow-up patients over five years to predict re-hospitalisation, but for this current report the mean duration of follow-up is not mentioned. Neither are any data provided about the prevalence of readmission during the follow-up period. Were all readmissions related to heart failure, and how many occurred? What were the factors identified by the ML algorithms to have predictive value? It may not be possible to give that information for the neural network analysis, but with the other methods some information should be available. How did a conventional logistic regression perform in this study and were the ML methods better?
The comparison of different methods of ML is useful, and the conclusion that machine learning methods were more accurate if they did not impute missing data is interesting, but it is difficult to interpret without more information about how much data were missing. In the 71% of subjects with some missing data, how many values were missing? It would of course be unsurprising for imputation to be inaccurate if >50% of values are missing.
Table 1 makes reference to clinical examinations but it is unexplained whether they were planned or haphazard. The abbreviation “IMA” is used, presumably for acute myocardial infarction; the abbreviation should not be used or it should be explained in full because this is an English transliteration of the Italian, rather than the standard English abbreviation (AMI for acute myocardial infarction; IMA is used for the internal mammary artery).
Author Response
1) The investigators compared different methods of machine learning to analyse data collected routinely from hospital and community-care settings, concerning patients diagnosed to have heart failure, for their predictive power to identify patients who will be readmitted. The number of subjects is relatively small for this type of study, at 388. Some data were missing in 71% of patients, so for some analyses only those patients with full data (N=110) were used. Perhaps to compensate for the limited number, analyses using the best performing method (GLMNet) were repeated 10,000 times, with a bootstrapping method of resampling, and then mean values for all the iterations were reported. Obviously, that risks overfitting, as the authors identify, and they do not report the performance of their method in a retest population that is independent.
At the beginning of the study a power analysis with a simulation-based approach was run to evaluate the minimum number of records to properly train the MLTs. Further details on the power analysis, that were not included in the original version of the manuscript, has been added in subsection 2.3.
2) The main concerns are that the paper misses important details. There is no information about how heart failure was diagnosed, and whether the population includes both heart failure with reduced and with preserved ejection fraction.
The diagnosis of heart failure was made according to the criteria of the European Society of Cardiology (ESC) using patient’s clinical history, physical examination, and data obtained from clinical tests (electrocardiogram, echocardiography, N-terminal pro-BNP (NT-proBNP)). For what concerns ejection fraction (EF), both patients with reduced and preserved EF were included in the study and were distributed as follows: 21% (reduced EF), 55% (mid-range EF), 24% (preserved EF). The information was added to the manuscript.
3) It is implied that the aim of the study is to follow-up patients over five years to predict re-hospitalisation, but for this current report the mean duration of follow-up is not mentioned. Neither are any data provided about the prevalence of readmission during the follow-up period. Were all readmissions related to heart failure, and how many occurred?
The study aimed to predict the first hospital admission of the patients after the enrollment in the GISC project. So that, we the aim of the study was not to predict hospital re-admissions but hospital admissions. The hospital admissions could be related or not to heart failure. The median duration of the follow-up has been added to the results section.
4) What were the factors identified by the ML algorithms to have predictive value? It may not be possible to give that information for the neural network analysis, but with the other methods some information should be available.
Further details on the predictive value of each predictor has been added in the Results section and in the new Table 6. Information on the important predictors was added for the MLTs for which it was possible to identify which covariate has an impact on identifying patients with at least a hospital admission.
5) How did a conventional logistic regression perform in this study and were the ML methods better?
The details about the performance of the conventional logistic regression model were reported in the results section.
6) The comparison of different methods of ML is useful, and the conclusion that machine learning methods were more accurate if they did not impute missing data is interesting, but it is difficult to interpret without more information about how much data were missing. In the 71% of subjects with some missing data, how many values were missing? It would of course be unsurprising for imputation to be inaccurate if >50% of values are missing.
Further details on the missing values have been added in the Results section.
7) Table 1 makes reference to clinical examinations but it is unexplained whether they were planned or haphazard. The abbreviation “IMA” is used, presumably for acute myocardial infarction; the abbreviation should not be used or it should be explained in full because this is an English transliteration of the Italian, rather than the standard English abbreviation (AMI for acute myocardial infarction; IMA is used for the internal mammary artery).
We would like to thank the reviewer for the comment. The abbreviation (IMA) was wrong. The correct one is AMI (Acute Myocardial Infarction). The manuscript was amended accordingly.
Round 2
Reviewer 2 Report
The authors have provided answers to most of the questions raised in the previous review, but they have not yet incorporated all those answers into the manuscript itself. For example, the proportions of subjects with heart failure with reduced ejection fraction, heart failure with mid-range ejection fraction, and heart failure with preserved ejection fraction, appear to be listed only in the response to the reviewer. They should be included in the manuscript itself. At line 108, for example, heart failure with mid-range ejection fraction is not even mentioned although it was by far the largest group. At line 104 the authors refer to an “informative database” but those results are not included in the paper or in any supplementary material.
To understand better the prognostic value of the machine learning techniques, it would be helpful to know more about the functional limitation of the subjects. Table 1 suggests that 2/3 were in NYHA class III but that is surprising given their ejection fractions – were they limited by dyspnoea from other causes too, or by ischaemia?
The authors have clarified that their study was designed to predict the first hospitalization (not rehospitalization). However, although it was requested, the causes of hospitalization are not given. The study has documented and not discriminated between all hospitalizations, which may not be helpful as it is likely to include many episodes that are not related specifically to heart failure itself.
It is now clear that the proportion of subjects with ischaemic heart failure is high and that there is also a high prevalence in this study population of patients with chronic obstructive pulmonary disease. The specific predictors that are now included (in new table 6) include acute myocardial infarction, heart failure with an ischaemic aetiology, chronic obstructive pulmonary disease, and co-morbidities. These results all suggest that admissions may have been caused by exacerbations of COPD, or by episodes of myocardial ischaemia, or perhaps even elective admissions for coronary arteriography. This makes the generalisability of the results debatable, certainly without more information.
In the same context it is surprising to read that brain naturetic peptide levels were only identified as predictors by two of the ML methods, and NYHA class was identified only by one.
With the exception of GLMN, the diagnostic performance of the learning methods was quite poor, usually with areas under the curve within the range of 60-70%. These results imply that the methods would not be useful or accurate enough to be applied in routine clinical practice. Most of the methods were no better than a standard logistic regression.
No information has been provided in response to the question about frequency of follow-up. It is presumed that that must have been opportunistic rather than standardised within this study.
At line 262 there is a word missing; the text should read … “was run to evaluate”..
At line 282, there is no verb in this sentence. It should read.. “BNP had percentages”.
The abbreviation for acute myocardial infarction has been changed to AMI, which is appropriate, but the meaning of this abbreviation should also be explained in a footnote to the figures in which it is used.
Author Response
The authors have provided answers to most of the questions raised in the previous review, but they have not yet incorporated all those answers into the manuscript itself. For example, the proportions of subjects with heart failure with reduced ejection fraction, heart failure with mid-range ejection fraction, and heart failure with preserved ejection fraction, appear to be listed only in the response to the reviewer. They should be included in the manuscript itself. At line 108, for example, heart failure with mid-range ejection fraction is not even mentioned although it was by far the largest group. At line 104 the authors refer to an “informative database” but those results are not included in the paper or in any supplementary material.
Done, the information about the distribution of the patients has been added to the manuscript.
To understand better the prognostic value of the machine learning techniques, it would be helpful to know more about the functional limitation of the subjects. Table 1 suggests that 2/3 were in NYHA class III but that is surprising given their ejection fractions – were they limited by dyspnoea from other causes too, or by ischaemia?
They were limited by dyspnoea. In the sample there was a high proportion of COPD patients, especially among those hospitalized. We included such consideration in the results.
The authors have clarified that their study was designed to predict the first hospitalization (not rehospitalization). However, although it was requested, the causes of hospitalization are not given. The study has documented and not discriminated between all hospitalizations, which may not be helpful as it is likely to include many episodes that are not related specifically to heart failure itself. It is now clear that the proportion of subjects with ischaemic heart failure is high and that there is also a high prevalence in this study population of patients with chronic obstructive pulmonary disease. The specific predictors that are now included (in new table 6) include acute myocardial infarction, heart failure with an ischaemic aetiology, chronic obstructive pulmonary disease, and co-morbidities. These results all suggest that admissions may have been caused by exacerbations of COPD, or by episodes of myocardial ischaemia, or perhaps even elective admissions for coronary arteriography. This makes the generalisability of the results debatable, certainly without more information.
The distribution of the hospitalizations according to the cause of hospital admission was as follows: 84.76% HF,15.24% other causes. Most of the hospitalizations were HF-related. We included such consideration in the results.
In the same context it is surprising to read that brain naturetic peptide levels were only identified as predictors by two of the ML methods, and NYHA class was identified only by one.
Such results could be related to the sample characteristics -which were homogeneous in terms of NYHA class since about 2/3 of the patients had a NYHA class of 3- or to the sample size which could be not enough to identify such characteristics as significant predictors of hospitalization. We included such consideration in the discussion.
With the exception of GLMN, the diagnostic performance of the learning methods was quite poor, usually with areas under the curve within the range of 60-70%. These results imply that the methods would not be useful or accurate enough to be applied in routine clinical practice. Most of the methods were no better than a standard logistic regression.
We agree with the reviewer that the performance of most of the methods employed to predict hospitalizations was not better than the logistic regression in this study. However, we cannot make conclusions about the usefulness of such methods in developing predictive models using clinical data. We cannot rule out that using a larger database and/or more detailed clinical information would improve the predictive performance, especially if the clinical data employed to predict the hospitalizations are characterized by complex relationships and non-linear interactions. Nevertheless, in such settings and the limited information available, it is not uncommon to don’t observe an overperformance of logistic regression as compared to other ML (https://www.ncbi.nlm.nih.gov/pubmed/20703563 ). We included such consideration in the discussion.
No information has been provided in response to the question about frequency of follow-up. It is presumed that that must have been opportunistic rather than standardised within this study.
No standard follow-up schedule has been foreseen in this study. In general, the follow-up was at least once a year, but, depending on clinical conditions and medical judgement, it could be more frequent. We included such consideration in the paper.
At line 262 there is a word missing; the text should read … “was run to evaluate”..
Done
At line 282, there is no verb in this sentence. It should read.. “BNP had percentages”.
Done
The abbreviation for acute myocardial infarction has been changed to AMI, which is appropriate, but the meaning of this abbreviation should also be explained in a footnote to the figures in which it is used.
Done
This manuscript is a resubmission of an earlier submission. The following is a list of the peer review reports and author responses from that submission.
Round 1
Reviewer 1 Report
The authors present an interesting work. The work is well presented. My major concern is that they do not present similar work presented in the literature and they do not compare their work with these methods.
Reviewer 2 Report
The paper is an interesting case study, aiming to predict hospitalization among patients with heart failure, using real data from the GISC study.
However, I have some concerns, summarized below:
- the paper doesn't study related work.
- the main argument of the authors of picking GLMNet, MAXENT and LogitBoost are that they are methods suitable to handle large datasets. The dataset they are using doesn't seem that big, and no comparison with other literature methods is performed.
- I believe the presentation (and the structure in some parts) of the paper can be improved for the content to be more clear.
In more detail:
- Abstract:
* "in predicting hospitalization in patients": in->of
* "than the other two MLT techniques": MLT is not yet defined
* "with and estimated the accuracy of": fix
- Introduction:
* "the sudden development of medical science": I am not really sure if you mean here that medicine has recent developments or technology instead
* "a large amount of heterogeneous clinical data, the so called big data": perhaps link the term big data better to the large amount rather than the heterogeneous or the clinical aspect of the data
* "accurate description of the disease's evolution": perhaps accurate tracking
* "a lifelong disease management ... represents a sever public burden": citation?
* "its prevalence rises from 0.6 cases ...to 28 ...": add time interval from the reference (e.g., in 5 years)
* "(HF) has a severe impact on the burden of hospitalisations,.... organizations": I am not sure what exactly you mean here. Rephrase.
* "Identifying the HF patients at higher risk of hospital readmission....": are you trying to predict admissions or re-admissions?
* This section completely lacks citations of related literature. Have you looked at other papers that try to predict hospitalizations of patients with heart conditions? (e.g., https://www.ncbi.nlm.nih.gov/pmc/articles/PMC4314395/ for predicting admissions or https://jamanetwork.com/journals/jamacardiology/article-abstract/2572174 for predicting re-admissions)
- Materials and methods
* "Analyses were performed only on complete cases": define complete cases. What does it mean? No-missing data? In that case, did you try any techniques that deal with missing data to observe if it's better to throw away patients with missing info or to keep them?
* "Continuous data": I believe you mean here numerical instead of continuous. For example, age is a discrete variable, not a continuous one, but it takes numerical values. Also I would prefer if you could use a word like features/variables instead of data here.
* "ethology related to...": is this text? If so, what pre-processing did you do with text?
* "dichotomous data": binary-valued/binary/bi-valued
* "dichotomous label": same as above
* Some really important information about the dataset are not mentioned at all: how many features correspond to each patient after the pre-processing? how many patients correspond to each class (hospitalized vs non-hospitalized)? I believe the dataset description should be improved a lot and expanded to include further details.
* How did you incorporate time into the model?
* It is not clear to me whether patients that are included in the dataset have been previously hospitalized.
* I would like to see some expanded high-level description of the models. Also the references to the methods in the paper seem to be to specific toolboxes and not the methods. In terms of structuring, I would propose to include a subsection "Implementation details" where the specific toolboxes are listed.
* I believe it shouldn't take much space to define PPV, NPV, sensitivity, specificity, and accuracy, even though they are well-defined.
* A good idea to compare methods with one single number is the AUC (Area Under the ROC curve), complementing AC1
- Results
* 110 patients is not a large dataset
* In the results section, you report multiple tables that look at descriptive information about the dataset (sample features/characteristics of the dataset), not results of the study.
* I suggest the following material re-organization: in the Materials and Methods section to have a subsection devoted to the dataset where: a) you describe where you are getting them from (GISC study), b) what kinds of features it includes (numerical vs categorical), c) data pre-processing that you do, d) numbers about the dataset (# of patients, # of features, # of hospitalized, # of non-hospitalized), e) tables that describe the dataset. Then another subsection that describes the methods that you will use, a description of each method and assumptions/advantages/disadvantages. Last the Materials and Methods section can close with a last subsection where you describe how the evaluation is performed (splitting into training/testing, how many patients in training/validation/test set and evaluation metrics)
* Wasn't there some parameter tuning to be done? Where is it described?
* Table 3: in each line the sum of the records is 16. Shouldn't that be as large as the test set? Is the test set only 16? Can you make that clear in the description?
* I believe for this study to be sound, you need to compare with other methods and perform a more thorough analysis. Why didn't you experiment with Support Vector Machines, logistic regression, random forests, AdaBoost, classification trees, rule-based classifiers?
* How do you compare with the literature? E.g., with the two citations I suggested in the Introduction?
* Is there a medical baseline you can compare with?
* In the clinical domain, interpretability is a very critical aspect. How such a system you are proposing will be faced by the medical staff? Would they trust it? How can your results help in that?
Discussion
* "one added value of the present analysis is that ..": this could appear in the body of the paper, where you describe where you got the data from. It is not a discussion of the results.
* the discussion section of the paper, should discuss a bit more the results presented in the previous section